# Decreased R:FR Ratio in Incident White Light Affects the Composition of Barley Leaf Lipidome and Freezing Tolerance in a Temperature-Dependent Manner

**DOI:** 10.3390/ijms21207557

**Published:** 2020-10-13

**Authors:** Terézia Kovács, Mohamed Ahres, Tamás Pálmai, László Kovács, Matsuo Uemura, Cristina Crosatti, Gabor Galiba

**Affiliations:** 1Biological Research Centre, Institute of Plant Biology, H-6701 Szeged, Hungary; kovacs.laszlo@brc.hu; 2Department of Plant Biology, University of Szeged, 6720 Szeged, Hungary; 3Centre for Agricultural Research, Agricultural Institute, 2462 Martonvásár, Hungary; mohamed.ahres@atk.hu (M.A.); palmai.tamas@atk.hu (T.P.); galiba.gabor@atk.hu (G.G.); 4Festetics Doctoral School, Georgikon Campus, Szent István University, H-2100 Gödöllő, Hungary; 5Department of Plant-Bioscience, Faculty of Agriculture, Iwate University, Morioka 020-8550, Japan; uemura@iwate-u.ac.jp; 6CREA Research Centre for Genomics and Bioinformatics, Fiorenzuola d’Arda, 29017 San Protaso, Italy; cristina.crosatti@crea.gov.it

**Keywords:** barley, cold acclimation, freezing, lipidome, lipid signaling, light regulation, monogalactosyldiacylglycerol, phospholipid

## Abstract

In cereals, C-repeat binding factor genes have been defined as key components of the light quality-dependent regulation of frost tolerance by integrating phytochrome-mediated light and temperature signals. This study elucidates the differences in the lipid composition of barley leaves illuminated with white light or white light supplemented with far-red light at 5 or 15 °C. According to LC-MS analysis, far-red light supplementation increased the amount of monogalactosyldiacylglycerol species 36:6, 36:5, and 36:4 after 1 day at 5 °C, and 10 days at 15 °C resulted in a perturbed content of 38:6 species. Changes were observed in the levels of phosphatidylethanolamine, and phosphatidylserine under white light supplemented with far-red light illumination at 15 °C, whereas robust changes were observed in the amount of several phosphatidylserine species at 5 °C. At 15 °C, the amount of some phosphatidylglycerol species increased as a result of white light supplemented with far-red light illumination after 1 day. The ceramide (42:2)-3 content increased regardless of the temperature. The double-bond index of phosphatidylglycerol, phosphatidylserine, phosphatidylcholine ceramide together with total double-bond index changed when the plant was grown at 15 °C as a function of white light supplemented with far-red light. white light supplemented with far-red light increased the monogalactosyldiacylglycerol/diacylglycerol ratio as well. The gene expression changes are well correlated with the alterations in the lipidome.

## 1. Introduction

Cold stress is one of the major limiting environmental factors of plant growth [1]. Overwintering plants have high plasticity to adapt to cooling [2] at multiple points during their life cycle from seed germination to transition from vegetative to reproductive phase and flowering. The cold acclimation process is also controlled by the circadian clock, day length, and illumination depending on both light intensity and spectral composition [3,4,5,6].

Winter-hardy plants can improve their freezing tolerance during a 2–7 week-long cold acclimation period prior to the appearance of frost periods under natural conditions. The threshold temperature that triggers cold acclimation is significantly higher in the case of winter-hardy cultivars than for sensitive ones. Furthermore, the capability of winter-hardy plants to sense the low red to far-red ratio (R:FR) conditions in the twilight period significantly contributes to a quick pre-hardening process leading to increased freezing tolerance. Indeed, it was reported in *A. thaliana* and more recently in cereals that freezing tolerance was increased at non-acclimating temperatures (15–16 °C) when plants were illuminated by supplemented with far-red light (WL+FR) [7,8,9]. The same C-repeat binding factor (CBF) molecular pathway was found to connect photoperiodic and light quality regulation with frost tolerance in both *A. thaliana* and cereals. That process is mediated by phyB and phyA [7,8,9,10]. It is also well established that some of the CBF-genes are circadian regulated and their expression reaches a maximum after 6–12 h after dawn in wheat and barley, but the FR light causes earlier transcription (ZT4 to ZT8) [6,9]. Apart from the CBF regulon, FR light-induced activation of PhyA also induces ABA signaling and subsequently JA signaling. The ABA pathway is rather important, because low R:FR treatment did not confer cold tolerance in an ABA-deficient mutant in tomato [11]. These authors concluded that JA functioned downstream of ABA to activate the CBF pathway. Apart from this, ABA does have another important role, namely it is involved in the regulation of the unsaturated fatty acids biosynthesis in plants [12]. Accordingly, it was hypothesized that ABA may activate SnRK2 and FUS3 and suppress PhyB (for details see the very recent review by He et al. [13]). A cluster of *CBF* genes are located in the *frost resistance 2 locus (Fr-A2)* on the long arm of chromosome 5A [14,15,16] in wheat and barley, and therefore they were postulated to be responsible for freezing tolerance. Among them, *CBF14* has a major role in the expression of freezing tolerance, both in wheat and barley [17,18,19,20]. We found that WL+FR illumination enhances *CBF14* expression and increases frost tolerance at 15 °C in winter wheat and barley genotypes, but not in *T. monococcum* (einkorn), which has a relatively low freezing tolerance [8].

To elucidate the interaction between far-red (FR) light and cold treatment in the pre-hardening process, barley plantlets were grown under different light conditions with low, normal, and high light intensities at 5 and 15 °C, and the expression of the *HvCBF14* gene and two well-characterized members of the CBF-regulon, *HvCOR14b* and *HvDHN5*, were studied [21]. The supplemented FR illumination induced the expression of *HvCBF14* and of its target gene *HvCOR14b* at both temperatures, and the combined effects of cold and FR supplementation increased freezing tolerance in synergy [21]. Earlier experiments demonstrated the impact of complementary FR illumination on *CBF* gene expression [6,8,22] in different cereals, and the strongest impact of the FR treatment was found at 15 °C after 10 days, whereas at 5 °C after 7 days.

In the recent years, several lines of evidence proved that lipids not only provide structural bases for cell membranes and energy stocks for metabolism in plant cells, but they are also involved in the initiation of defense reactions, as signal mediators in the mitigation processes in response to stress [23,24]. For instance, under WL+FR conditions *Hordeum vulgare phosphatidylinositol 4-kinase (HvPI4K)* displayed an early phase of expression under light/dark cycles [6], and this signaling pathway plays a role in Ca^2+^ release from intracellular Ca^2+^ stores [25,26]. It is known that calmodulins bind Ca^2+^ when Ca^2+^ levels increase and activate the calmodulin-binding transcriptional activator (CAMTA) proteins. In *A. thaliana* CAMTA transcription factors bind to the conserved DNA motif 2 (CM 2) of the promoter of *CBF* genes. The calmodulin-binding transcriptional activator 3 (CAMTA 3) is a positive regulator of *CBF2* expression, and the double *camta1 camta3 A. thaliana* mutant showed impaired freezing tolerance [27]. In cereals, a direct connection between the regulating effect of CAMTA proteins and *CBF* expression has not been disclosed yet [6]. Currently, signaling lipids involved in sensing various stresses such as light and low-temperature are attracting more scientific attention, and among them phosphatidylinositol (PI) and sphingolipids (SL) (the common name is cerebrosides) are the main lipid species [28]. The fact that plant sphingolipids participate in cellular signaling and they are not only structural elements of membranes has also resulted in an increased interest in these molecules. Ceramides are one group of SLs composed of a long-chain base (LCB) bound to a fatty acyl chain via an amide bond. The hydroxylation and unsaturation level, as well as the fatty acyl chain length of ceramide constituents, is highly variable. In plants, SLs are synthesized through two pathways: a “de novo” and an alternative salvage route. Ceramides are synthesized by ceramide synthases in the endoplasmic reticulum, where they are further modified to provide more complex SLs [29]. About a thousand SL molecular species have been identified in plants, which makes SLs the most structurally diverse group of lipids [30,31]. Ceramides might play an important role as signaling lipids in both prokaryotes and eukaryotes [32,33,34,35,36]. In plants, SLs are important for plant development, adaptation to abiotic stress and the response to pathogens [37,38,39,40,41,42].

Earlier experiments have shown that membrane rigidity increases when plants are exposed to cold temperature, resulting in the distortion of the structure of membrane-embedded proteins including photosynthetic complexes [43]. The modification of membrane lipid composition can help to avoid cold-induced damage [2]. This process is strictly regulated [44] through the following traits: membrane fluidity/rigidity, fluctuation of glycerolipid levels [45], saturation level of lipids [46,47,48,49,50,51], and the ratio of the bilayer (BL) forming and non-bilayer-forming (NBL) lipids [52,53]. Previous studies have identified some key elements in the response to cold treatment of plants, such as MGDG/DGDG, phosphatidylcholine/phosphatidylethanolamine ratio (PC/PE), increase in PA [54], change in the amount and composition of SLs [55,56,57,58,59,60] and diacylglycerol acyltransferase signaling [61]. Furthermore, ceramides are involved in signal transduction that regulates the amount and composition of SLs in the membrane by cold treatment [33,38].

In our studies, we wanted to explore the role of membrane lipids in cold acclimation during the vegetative stage of cereals. Based on recent experiments, we studied the expression of the *monogalactosyl diacylglycerol synthase 2 (MGD2), digalactosyldiacyl glycerol synthase 1 (DGD1)*, and *neutral ceramidase (NC)* genes that may play significant roles in the regulation of lipid composition during cold treatment [62,63,64]. *MGD2* plays a role in the synthesis of major structural components of photosynthetic membranes and in the biogenesis of envelope. This enzyme can use both prokaryotic (18:1/16:0) and eukaryotic (18:2/18:2) 1,2-diacylglycerol species, but operates with some preference for the eukaryotic one. *MGD2* is also responsive to phosphate deficiency [63]. *DGD1* is involved in the biosynthesis of DGDG and is located in the thylakoid membrane. *DGD1* is specific for alpha-glycosidic linkages and is responsible for the final assembly of galactolipids in photosynthetic membranes [65]. The *NC* family consists of three members: *neutral ceramidase 1 (NC1)* serves to maintain SL homeostasis. It is involved in the control of oxidative stress. *Neutral ceramidase 2 (NC2)* hydrolyzes the SL ceramide into sphingosine and free fatty acid. *Neutral ceramidase 3 (NC3)* is involved in SL biosynthesis [62]. As under stress conditions the levels of phospholipase D and A are also elevated [54,66,67], the relative mRNA expression of *phospholipase D alpha 1 (PLDα1)*, *phospholipase D alpha 2 (PLDα2)* and *PLDα3 (phospholipase D alpha 3)* was also selected for investigation. Plant lipoxygenases may be involved in a number of diverse aspects of plant physiology including growth and development, pest resistance, and senescence or responses to wounding. These enzymes catalyze the hydroperoxidation of polyunsaturated fatty acids. Based on our previous results, the expression of the *linoleate 9S-lipoxygenase 2 (LLO2)* gene can be detected principally during transient increases in lipoxygenase activity in seedlings. Jasmonic acid (JA), derived from linolenic acid via lipoxygenation in plant cells is recognized as a growth regulator [68,69]. *Lipoxygenase 2.3 (LOC)* is required for the wound-induced synthesis of JA in leaves [70,71]. *Non-lysosomal glucosylceramidase (GC)* catalyzes the hydrolysis of glucosylceramide to free glucose and ceramide [28]. *Aldehyde reductase (ARL)* catalyzes the reduction of the aldehyde carbonyl groups on saturated and α, β-unsaturated aldehydes with more than five carbons. It may act as a short alcohol-polyol-sugar dehydrogenase possibly related to carbohydrate metabolism and the acquisition of desiccation tolerance. It may also be involved in signal transduction [72]. *Alcohol dehydrogenase (AD3)* provides a constant supply of NAD+. If the roots lack oxygen, the expression of *AD3* increases significantly. Its expression is also increased in response to dehydration and low temperatures; it plays an important role in fruit ripening, seedling and pollen development [73,74]. To elucidate the cold acclimation process of plants, lipidomic approaches have been extensively applied to the model plant *A. thaliana* but scarcely to winter cereals such as barley. Moreover, according to our best knowledge, the possible lipid background of the light quality induced pre-hardening process has not been studied yet. Lipids, especially PLs are the key components, at least as signal molecules, during the phyA-regulated acclimation process [6,7]. It is important to emphasize that phyA is the primary plant photoreceptor responsible for mediating photomorphogenic responses in FR light, and is essential for survival in canopy shade [75,76,77]. Consequently, its involvement in the pre-hardening process can be considered as a temperature-dependent pleiotropic effect.

Our main goal was to elucidate in what way the plant lipidome responds to FR light, which eventually leads to a significant improvement in the freezing tolerance of plants. Applying the most up-to-date lipidomic analysis technique (electrospray ionization triple quadrupole mass spectrometry [78]), we could determine how the membrane lipid species are quantitatively and qualitatively altered under different temperature and light conditions. This study demonstrates that WL+FR can substitute cold treatment at higher temperature (15 °C), and mostly acts in a synergistic manner at cold (5 °C) nonfreezing temperature.

## 2. Results

### 2.1. Addition of FR Light to WL in the Incident Light Improves Basal Freezing Tolerance in a Temperature-Dependent Manner

Plants grown either at 5 or 15 °C under 12-h light/12-h dark photoperiods were exposed to white light (WL) (control) and WL+FR and tested for their freezing tolerance. The results of the electrolyte leakage measurements showed clear differences between the light treatments (Figure 1). At 15 °C, a significant decrease was observed in the relative conductance level of the WL+FR treatment, but only at −7 °C freezing temperature. When the plants were exposed to 5 °C, a substantial reduction in electrolyte leakage was observed, and after freezing at −10 °C the deviation was almost 20% between the different light treatments, whereas at −12 °C this diversion was even greater (22%). These results indicate that WL+FR positively affected the freezing tolerance of barley plants in a temperature-dependent manner.

### 2.2. Expression Analysis

The expression of *MGD2* mRNAs in the sample grown under WL+FR was significantly greater (*p* < 0.05) at 15 °C after 1 day (Figure 2A), but the differences did not persist on day 10 since the mRNA level also increased in the WL control. When the temperature was shifted from 15 to 5 °C, the expression of *MGD2* was significantly lower (*p* < 0.05), after 1 day under WL+FR illumination (Figure 2C).

The expression of *DGD1* was significantly greater (*p* < 0.005) following exposure to WL+FR after both 1 (Figure 2A) and 10 days at 15 °C (Figure 2B). Compared to the control WL, the mRNA level of *DGD1* was lower as a consequence of FR supplementation after 1 day at 5 °C (Figure 2C) and was drastically lower, independently of the spectral composition of the incident light, after 7 days at 5 °C (Figure 2D).

The *NC* expression profile was very similar to that of *DGD1* at 15 °C (Figure 2A,B), whereas no variation in *NC* mRNA level was detected at 5 °C in response to a change in WL+FR after 1 day (Figure 2C); then, after 7 days, the gene was expressed at a higher level in the sample exposed to WL+FR compared to WL, mainly due to down-regulation of *DGD1* mRNA in the latter (Figure 2D).

The expression level of the *PLDα1* and *PLDα3* genes turned out to be temperature-sensitive (Figure 2); after 1 day, it was moderately lowered but a more significant reduction in expression was detected after 7 days at 5 °C. Although the same expression trends could be observed in both WL and WL+FR light regimes, FR light illumination alleviated the cold temperature repression at day 7. WL+FR slightly increased the expression of both genes after 7 days at 5 °C (Figure 2D). Notably, the expression of *PLDα2* was hardly detectable.

WL+FR significantly (*p* < 0.05) increased the expression of *LLO2* after 1 day at 15 °C (Figure 3A), whereas after 10 days variation of WL+FR did not impact significantly on the expression of *LLO2* (Figure 3B). When the same light conditions were tested in plants exposed to 5 °C, this expression pattern was reversed and WL+FR illumination significantly (*p* < 0.05) lowered the expression of *LLO2* after both 1 and 7 days (Figure 3C,D). The gene expression profiles of *LOC* and *AD3* were very similar to that of *LLO2* described above (Figure 3), except that at 5 °C the *LOC* mRNA level was not greater after 7 days as a result of FR enrichment.

WL+FR elevated the expression level of *GC* and *ARL* only after 10 days at 15 °C compared to the corresponding WL-grown sample (Figure 3A,B). At 5 °C, this WL+FR treatment induced a significantly greater expression of GC (*p* < 0.05) only after 1 day, whereas no effect was detected after 7 days (Figure 3C,D).

Summary diagrams of the gene expression results can be found in Appendix A. According to the hierarchical clustering of gene expression levels (Appendix A), the most characteristic gene expression changes were observed as a function of WL+FR illumination after 7 days at 5 °C, showing the synergistic relationship between the two treatments.

### 2.3. Mass Spectrometry Analyses of Isolated Lipids

For lipid analyses, total lipid extracts from barley leaves were used. Electrospray ionization triple quadrupole mass spectrometry was employed to monitor lipid changes that occur both during WL+FR and cold treatment. The alterations caused by WL+FR in membrane lipid quantity and quality were compared at 5 °C and 15 °C.

#### 2.3.1. Changes in Lipid Class Distribution

The effect of FR and 5 °C on lipid class distribution was determined by LC-MS/MS. Table 1 shows the changes in total lipid contents. WL+FR illumination slightly (not significantly) increased the total lipid content in barley leaves grown at 15 °C. Cold treatment (5 °C) significantly reduced total lipid content after both 1 and 7 days in the leaves of the control WL-illuminated plants. Total lipid content in the WL+FR-treated plants remained constant during the cold treatment (Table 1).

Most of the lipids identified in barley leaves illuminated with both WL and WL+FR belong to the MGDG lipid class (Figure 4). Upon WL+FR treatment, the MGDG content in barley leaves was significantly greater at 5 °C after 1 day. DGDG content was not altered substantially (Figure 4A’–D’). As for the other lipid classes, only PG, PE, and PS responded to WL+FR significantly. Compared to WL, WL+FR treatment resulted in greater PS content in the leaves, reached at different rates but significantly under each set of experimental conditions.

PE content was significantly greater after 1 day of WL+FR treatment at 5 °C, whereas PG increased significantly both after 1 day at 5 °C and after 10 days at 15 °C (Figure 4A–C).

The ratio of MGDG and DGDG, the main lipid components in the chloroplast that form the double-layered membrane was greater as a result of FR treatment at each sampling point, most substantially at 5 °C after 1 day (Table 2).

The PC to PE ratio determines the membrane chilling responses, and moreover, serves as an index for membrane resistance against frost injury [79]. In general, PC/PE was lower in the leaf samples as a consequence of WL+FR. Changes induced by WL+FR treatment were in synergy with cold treatment, since after lowering the temperature from 15 to 5 °C, PC/PE decreased even in the WL-illuminated control (Appendix A).

#### 2.3.2. Lipid Species Distribution within Different Lipid Classes

Although there was no significant alteration within the MGDG lipid molecular class in barley leaves upon WL+FR treatment at 15 °C (Appendix A), an increase in the amount of each lipid species was noticeable at 5 °C (Figure 5A,B). In the case of the DGDG molecular species, WL+FR increased the content of 34:2, 36:5 and 34: 2, 36: 3 lipid species in the barley leaves at 5 °C after the 1 and 7-day treatment (Figure 5C,D).

The total HexCer content increased significantly (*p* < 0.05) from 215 nmol/mg dry weight to 295 in barley leaves upon WL+FR treatment at 15 °C both in 1 and 10 days in comparison with WL-illuminated samples. Among the different HexCer lipid molecular species, the highest increment was observed in the amount of the (42:2)-3 lipid molecular species at WL+FR and 15 °C. This deviation was highly significant (*p* < 0.01) after 10 days (Figure 6B). The total HexCer content of barley leaf samples illuminated by the control WL did not change significantly under treatment at 5 °C. Contrarily, when the plants were grown under WL+FR illumination, the ratio of HexCer (42:2)-3 increased significantly in plants kept for 7 days at 5 °C (Figure 6D). This lipid molecule contains two DBs; consequently, its increased amount could be important in the cold acclimation process, because it might increase membrane fluidity [66].

The concentration of each PA species, especially the 34:1 (*p* < 0.01) and 36: 4 (*p* < 0.01), 36:3 (*p* < 0.01), 36:2 (*p* < 0.01) polyunsaturated lipids, increased in barley leaves as a result of WL+FR treatment at 5 °C at both sampling dates (data not shown). The measured small alterations were not expected, because PA is an important precursor for lipid biosynthesis and plays a role in signaling pathways [54].

#### 2.3.3. Changes in the Unsaturation Level of Fatty Acids

To review cell membrane lipid unsaturation, the double-bond index (DBI) was calculated based on the percentage of lipid species and total double bonds (DB) identified by lipidomics. Table 3 shows the DBI of total lipids of plants treated at different temperatures. Unexpectedly, we could not observe significant differences in DBI under any conditions we applied [49]. However, we found significant differences in certain lipid classes.

The saturation of PE species can be seen in Table 4. With WL+FR illumination, the level of unsaturation was greater on day 1 at 15 °C, and this trend continued until day 10 of the treatment. The same trend can be seen on days 1 and 7 at 5 °C. At 15 °C, there is a significant difference in the number of species containing 2 and 5 DBs and 3, 6 and 5 DBs on days 1 and 10, respectively. Saturation levels were significantly greater on day 1 for the 5 °C treatment to 3, 2, 5 and 4, but this difference diminished by day 7 of the treatment (Table 4).

#### 2.3.4. Changes in the Lysophospholipid Level of Fatty Acids

LPD serves as a precursor pool available for PC and PE biosynthesis [54]. These molecules are also considered as indicators for the correctness of lipid extraction from various samples. Upon decreasing the temperature from 15 to 5 °C, the amount of LysoPC was lower in the WL-illuminated control samples. Contrary to this phenomenon, WL+FR illumination increased their concentration substantially upon a shift to low temperature (Table 5). However, LysoPE content was altered as a consequence of temperature decrease; moreover, WL+FR illumination does affect its concentration in the leaf samples (Table 6). From a technical point of view, we can conclude that the low amount of LysoPC lipids in the samples indicates that our extraction procedure was appropriate (see Materials and Methods).

## 3. Discussion

It is well established that illumination affects several aspects of plant development including stem elongation, shoot branching and flower induction. In addition, more and more evidence has been accumulated recently about the active adaptation of plants to the current and impending changes in the environment by using light-derived non-photogenic signals coming from photosensory receptors [80,81]. Several studies in *A. thaliana* and cereals showed the modulation of cold acclimation by phytochrome-derived signals affecting the expression of the CBF regulon [6,7,8,82]. The changes in the lipidome that maintains membrane integrity during frost as a result of FR-induced freezing tolerance in plants have not been described yet. Therefore, we set out to study the alterations in the composition and quality of membrane lipids and the expression of possible related genes in barley leaves subjected to a combination of temperature and WL+FR light treatments. WL+FR illumination markedly elevated the PG and PS levels of barley leaves after 10 days at 15 °C as compared to the WL control, whereas the levels of other lipid classes remained unchanged (Figure 4B). Intriguingly, WL+FR caused bigger alterations in the lipid composition at 5 °C than at 15 °C. At 5 °C, WL+FR illumination led to markedly increased levels of MGDG, PG, PE, PI and PS already after 1 day (Figure 4C). PE, PS, and PG are prominent lipid classes, which have important roles in signal transduction processes [80]. We presume that elevated amount of PG (Figure 4) promotes the proper functioning of the thylakoid membrane at 5 °C [54]. PG is known to play a significant role in the construction of the thylakoid membrane and is an important constituent of photosynthetic complexes [83].

It is generally recognized that the unsaturation of fatty acid side chains plays an important role in cold acclimation by modulating membrane fluidity [43]. The increase of DBI would imply more fluid membrane composition. Although we could not observe any significant alterations in total lipid DBI (Table 3), some lipid classes showed a significant increase in DBI. For example, WL+FR illumination elevated the amounts of the 34:2 and 36:3 DGDG molecular species, which were significantly greater after a 7-day WL+FR treatment (Figure 5D). Under WL+FR illumination, the proportion of the HexCer (42:2)-3 lipid molecular species was significantly greater after 10 days at 15 °C (Figure 6B). Furthermore, the ratio of the HexCer (42:2)-3 molecular species was also significantly greater among ceramides in plants kept at 5 °C for 7 days (Figure 6C). We may consider these alterations as a cold acclimation process triggered by WL+FR illumination.

Previous research has shown that MGDG and DGDG account for the largest proportion of the total lipid composition of barley chloroplast membranes [84]. In addition, an increased ratio of BL to NBL membrane lipids (i.e., a decrease in MGDG/DGDG) was observed in plants responding to low temperatures [52,54,85]. Our experimental results contradict this trend to some extent, because as a result of WL+FR treatment the MGDG/DGDG ratio increased at both 5 and 15 °C (Table 2). According to our frost test, the small increase in the MGDG/DGDG ratio does not have any detrimental effect on membrane stability in barley leaves, suggesting that the other changes triggered by WL+FR treatment, either in lipid composition or DBI content, counterbalanced this modification. Notably, membrane fluidity increases parallel to the ratio of phospholipids/free sterols in plants, which improves cold tolerance [86].

It was reported that the total amount of PC and PE increased during cold acclimation in rye, *A. thaliana* and wheat leaves, and the gradual increase in PE parallels the cold-induced increase in frost tolerance [86,87,88]. Under our experimental conditions, the 7-day cold treatment at 5 °C did not significantly affect the total amount of PC and PE (Figure 4D). During this treatment, the plants could reach only a pre-hardened phase, because full cold hardiness in winter cereals is achieved after 3 to 5 weeks of cold treatment at −2–4 °C [88,89]. Surprisingly, however, the amount of PE increased as a result of WL+FR illumination after 1 day at 5 °C (Figure 4C). Therefore, we presume that the applied FR light accelerated the cold acclimation process. The other, and probably more important, alteration caused by WL+FR treatment was the significant increment in the number of DBs in the different molecular classes belonging to PE (Table 4). The ratio of the increment within the PE molecular species was much more pronounced. The amount of unsaturated molecular species in the plasma membranes has been correlated with cold-acclimation in cereals and *A. thaliana* [86,87,90,91,92]. The frost-resistant “Onice” winter barley, for example, had a much higher free fatty acid content (predominantly unsaturated) than the frost-sensitive barley cultivar “Gitane” [89]. Accordingly, Bohn et al. (2007) [88] suggested that the changes in PE molecular species during cold acclimation can be used as a diagnostic marker to predict the degree of frost tolerance in cereals. Based on our results, we highly agree with this statement and presume that WL+FR illumination increased freezing tolerance of the winter-hardy barley cultivar “Nure” at 15 °C through a modification of the abundance of different PE molecular species. The above results also warrant future studies to elucidate the role of barley fatty acid desaturase and SL desaturase enzymes in cold acclimation.

Changes in membrane lipids under cold or freezing stress influence the formation of lipid signaling molecules. Cellular receptors recognize lipid molecules by their phosphate head groups [93]. The levels of lysophospholipids species, LysoPC, LysoPE, and PA increased when *A. thaliana* plants were exposed to cold treatments [54]. The PA level was also greater in cold-tolerant *Eutrema salsugineum* plants slightly similar to the “Nure” cultivar of barley [94]. In our experiments, higher total LysoPC (Table 5) and lower total LysoPE (Table 6) levels were detected in leaves exposed to WL+FR light for 10 days at 15 °C, but the PA level did not change. This tendency was more pronounced for plants grown at 5 °C (Table 5 and Table 6) under the same illumination. The PA levels were also raised higher at 5 °C by WL+FR illumination as early as on day 1 of the treatment. These results may indicate that WL+FR illumination may induce a signal transduction pathway, thus increasing the viability of the plant grown at 5 °C as compared to WL only.

The HexCer lipid family is known to be involved in a variety of stress processes and cell death, as well as in ABA signaling activity [73,74]. However, while the level of the HexCer species (42:2)-3 did not change significantly either at 5 or at 15 °C after 1 day (Figure 6A,C), the amount of this HexCer species markedly increased after day 10 at 15 °C (Figure 6B) and particularly after day 7 at 5 °C (Figure 6D). The most significant difference was on day 7 at 5 °C. This lipid species may be involved in the signal transduction induced by WL+FR. We also investigated the related gene expressions. Expression of *NC* was strongly up-regulated by WL+FR light after 1 day at 15 °C (Figure 2A). The decreased FR ratio has an effect on the expression of *MGD2* and *DGD1* after 10 days at 15 °C and after 7 days at 5 °C, although only the *DGD1* gene expression was significantly greater (*p* < 0.01) (Figure 2). Interestingly, the MGDG and DGDG content did not correlate with the expression of *MGD2* and *DGD1.* It was reported that red and FR light both induce an increase in *MGD2* mRNA and enzyme activity in etiolated cucumber cotyledons, suggesting a role for the photoreceptor PhyA . PhyA was also reported to regulate the FR-induced upregulation of the CBF regulon responsible for frost tolerance in both *A. thaliana* and cereals [30,88,95].

It has been shown that low temperature activates phospholipase A and phospholipase D in *A. thaliana* [54]. It was also reported that short-term cold treatment activates phospholipase D and phospholipase A with galactolipid substrates [96,97]. This process can lead to the destabilization of membrane BLs, membrane fusion and cell death [54,98]. In our experiments, WL+FR illumination lowers the expression of *PLDα3* after 1 day at 5 °C (Figure 2C), suggesting that WL+FR illumination alleviates the effect of 5 °C. As a result, the amount of NBL lipids was greater by WL+FR illumination.

The relative expression of *LOC* (Figure 3A) was down-regulated by WL+FR illumination at 15 °C. In contrast, a greater relative expression of *LOC* was found at 5 °C when illuminated by WL+FR (Figure 3C). Similar changes occurred in the relative expression of *LLO2* mRNA (Figure 3C), which was more pronounced after 7 days at 5 °C. Lipoxygenase enzymes participate in oxylipin signaling. Fatty acid hydroperoxides produced by 9- or 13-lipoxygenases can be further converted enzymatically to various oxylipins including, most notably, JA. Thus, the activity of 13-lipoxygenases is linked to JA-mediated cold-stress tolerance [99].

The stronger response during 5 °C treatment is also supported by the up-regulated expression of *AD3* (Figure 3C). Accordingly, a similar trend could be observed in the relative expression of *GC*, but only after 10 days at 15 °C (Figure 3B) and after 1 day at 5 °C (Figure 3C).

The present findings confirm that WL+FR affected both lipid content and quality in barley leaves (Figure 7). Opposite to the expected effect of cold treatment, MGDG/DGDG was greater under WL+FR illumination in all cases. WL+FR illumination elevated the amount of PE and also significantly increased the ratio of the molecular species with high DB content that belong to both the PC and PE lipid classes. Similarly to this phenomenon, the LysoPC content also changed in opposite directions during the two treatments. The expression of *MGD2, DGD1, NC, LLO2, LOC*, and *AD3* genes was markedly up-regulated as a consequence of FR light enrichment at 15 °C after 1 day. We suggest that FR-induced freezing tolerance should be considered as an important part of the pre-hardening phase.

## 4. Materials and Methods

### 4.1. Strains and Growth Conditions

A *H. vulgare* ssp. vulgare winter barley genotype, cv. “Nure” with good cold tolerance was studied. Three days after seed germination, seedlings were planted into Jiffy-7 36 mm diameter feed disks (Jiffy Group, Oslo, Norway). Subsequently, plants were transferred to a PGV-36 (Conviron PGV36; Controlled Environments Ltd., Winnipeg, MB, Canada) growth chamber equipped with a modular LED light ceiling. The plants were grown at 15 °C day/night temperature, 12-h day length and illuminated by a continuous wide-spectrum LED (Philips Lumileds, San Jose, CA, USA, LXZ25790-y) WL at 250 photosynthetically active radiation (PAR) intensity (µmol m^−2^ s^−1^, (PPFD)) for 14 days. The plants received 50% Hoagland medium 3 times a week.

### 4.2. Light and Temperature Treatments

Half of the two-week-old plantlets were transferred to another growth chamber equipped with the same type of modular LED light ceiling as described above. Both chambers were divided into 2 areas. In the control area, the light conditions remained the same as during pre-nursing (12-h day length and 250 PAR WL). In the other area, supplementary FR illumination was added to WL with a narrow 750 nm LED (Edison Edixeon, 2ER101FX00000001). The control WL did not contain any FR. The R:FR reached 0.5 as a result of FR supplementation (Appendix A). The total light intensity remained the same 250 PAR as in the control WL. The temperature was the only difference between the two chambers: 15 °C in the first chamber and 5 °C in the second one. Samples were taken on the 1st and 7th days from plants kept at 5 °C and on the 1st and 10th days from plants kept at 15 °C. In both cases, lighting was switched on at 06:00 h, and the samples were collected between 12 and 14:00 (i.e., after 6–8 h illumination).

### 4.3. Freezing Test of Leaf Samples

To evaluate the frost tolerance of the plants, we measured the electrolyte leakage levels, as previously described [100], using small leaf samples frozen in a GP200-R4 liquid freezing system (Grant Instruments, Shepreth, UK) in five biological repetitions per treatment per freezing point. For plants grown at 15 °C, the freezing temperatures were −5, −7, and −9 °C, whereas leaf samples of plants exposed to 5 °C were frozen at −8, −10, and −12 °C. The samples were shaken for 2 h in MQ water and then the determination of the electrolyte leakage level was carried out with a Conductometer (Mikro KKT, Budapest, Hungary). Data analysis was performed with Multi-Sample Conductometer version 1.0 (Intron Software, Biological Research Centre, Szeged, Hungary (Copyright© L. Mencel 2002)).

### 4.4. qPCR Measurement

Total RNA was isolated from leaf samples with the Direct-zol™ RNA MiniPrep kit (Zymo Research Corp., Irvine, CA, USA) according to the manufacturer’s instructions. Subsequently, cDNA libraries were made using Moloney Murine Leukemia Virus (M-MLV) Reverse Transcriptase and oligo (dT) 18 primers (Promega Corporation, Madison, WI, USA). Relative expression levels were determined using KAPA SYBR^®^ FAST, Master Mix (2X), Universal qPCR kit (Kapa Biosystems, Inc., Wilmington, MA, USA) with CFX96 Touch™ Real Time PCR Detection System (Bio-Rad Hungary Ltd., Budapest, Hungary). PCR primers were designed using the NCBI-Primer Design Tool (National Center for Biotechnology Information, Bethesda, MD, USA) software. Relative expression levels were calculated by the ΔΔCt method, where cyclophilin was used as the reference gene [101]. The primer sequences and references are available at Appendix A [102]. The already known synthetic pathways were used for processing the results of the qPCR measurement. Before processing, we selected enzymes that may play a significant role in the regulation of the pathways we studied. The following databases were used to identify these genes and enzymes: Arabidopsis Acyl-Lipid Metabolism Pathways (http://aralip.plantbiology.msu.edu/pathways/phospholipid_signaling), Tair (https://www.arabidopsis.org/), ExPasy (https://enzyme.expasy.org/) The lipid results that we obtained in the ESI-MS/MS lipid profile measurement also helped us to select the appropriate gene.

### 4.5. Lipid Isolation and Lipidomics

Total lipids were extracted from intact barley leaves according to [54] with small modifications. The fresh weight of each leaf sample was 0.2 g. For lipid extraction, we collected the leaves separately from three plants from each temperature regime on the 1st and 7th or 10th day of treatments. At sampling, the middle third of flag leaves was cut and chopped and put instantly into a 10-mL glass tube with a Teflon-lined screw-cap (Thermo Fisher Scientific, Inc., Waltham, MA, USA) containing 3 mL isopropanol/0.01% BHT mixture at 75 °C and were incubated for 30 min. Next, we added 1.5 mL chloroform and 0.6 mL water to the solvent and agitated the sample at room temperature for 1 h, then we added 4 mL chloroform/methanol (2:1)/0.01% BHT extraction solvent, followed by shaking for 30 min. We collected the extract into a clean glass tube. We washed the barley leaves with extraction solvent five times; the last wash was 12 h long. We added 1.0 mL 1 M KCl to the pooled extracts. After vortexing, the phases were separated by centrifugation and the upper phase was discarded. Then we added 2.0 mL of water to the extract and repeated the phase separation. At this stage, the solvent was evaporated from the extract in a nitrogen evaporator. Finally, the lipid extract was dissolved in 1 mL chloroform and stored at −80 °C. To determine the dry weight, the extracted leaf pieces were dried at 110 °C in an oven overnight, cooled, and weighed. Dry weights were determined using a balance (Mettler Toledo AX, Mettler Toledo International, Inc., Columbus, OH, USA), which had a detection limit of 2 µg. The mass spectrometry analysis of the total lipid extracts was carried out by the MS-based method and performed at the Kansas Lipidomics Research Center Analytical Laboratory (https://www.k-state.edu/lipid/analytical_laboratory/lipid_profiling/index.html).

### 4.6. ESI-MS/MS Lipid Profiling

An automated electrospray ionization-tandem mass spectrometry approach was used, and data acquisition and analysis and acyl group identification were carried out as described [103].

Unfractionated lipid extracts were introduced by continuous infusion into the ESI source on a triple quadrupole MS/MS (API4000, ABSciex, Framingham, MA, USA). Data processing was described at the Kansas Lipidomics Research Center home page (https://www.k -state.edu/lipid/analytical_laboratory/lipid_profiling/index.html). The molecular species were identified by the presence of a specific head group fragment and the mass. The mass generally allows us to assign a specific number of acyl carbons and the number of DB. After correction for isotopic overlap, the peaks on the spectra were quantified in comparison to a group of internal standards. Generally, two internal standards of each head group class were used. The amounts were normalized as nmol/mg dry weight. One signal unit equals 1 nmol of internal standard from the same lipid group (usually with an adjustment for variation in response with m/z) (https://www.k-state.edu/lipid/analytical_laboratory/lipid_profiling/index.html). The precursor and neutral loss methodologies that are routinely utilized contrast with the more typically used product ion scans. The first mass spec corresponding to a particular molecular ion mass is held at a constant electrical field, while scanning is performed with the second mass spec to detect the fragments made in the collision cell. Product ion scans can be used to determine the head group or other fragments that are produced, as well as to identify the fatty acyl groups, in particular polar lipid molecular species, when this additional analysis is required.

### 4.7. Double Bond Index

The DBI of each lipid molecular species was calculated as the product of the amount of lipid molecular species and the average number of DBs/acyl chain, where the average number of DBs/acyl chain was calculated by dividing the number of DBs in the lipid molecular species by the number of acyl chains. Finally, the DBI of a lipid head group class was calculated as the sum of the unsaturation indices of individual lipid molecular species in that class according to the following equation [Σ(% of normalized signal intensity/mg dry weight of lipid species × no. of DB)]/100, as described in [43].

### 4.8. Statistics and Reproducibility

Three independent biological replicates were used at each measurement. The significance of the difference between WL+FR samples grown at 5 or 15 °C and the corresponding WL control samples were determined by Student’s *t*-tests using GraphPad Prism version 8.0.1 for Windows (GraphPad Software, San Diego, CA USA) software. Significance levels in figures are designated as * *p* < 0.1, ** *p* < 0.05, *** *p* < 0.01.

## Figures and Tables

**Figure 1 ijms-21-07557-f001:**
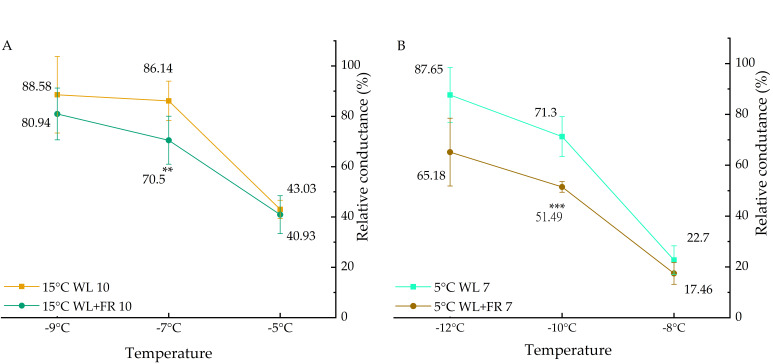
Relative conductance of barley leaf segments frozen at −5, −7 and −9 °C after incubation at 15 °C for 10 days (**A**) and frozen at −8, −10 and −12 °C after incubation at 5 °C for 7 days (**B**) under WL or WL+FR. Mean values ± SD of 5 independent biological repetitions are shown. Significance levels are designated as ** *p* < 0.05, *** *p* < 0.01.

**Figure 2 ijms-21-07557-f002:**
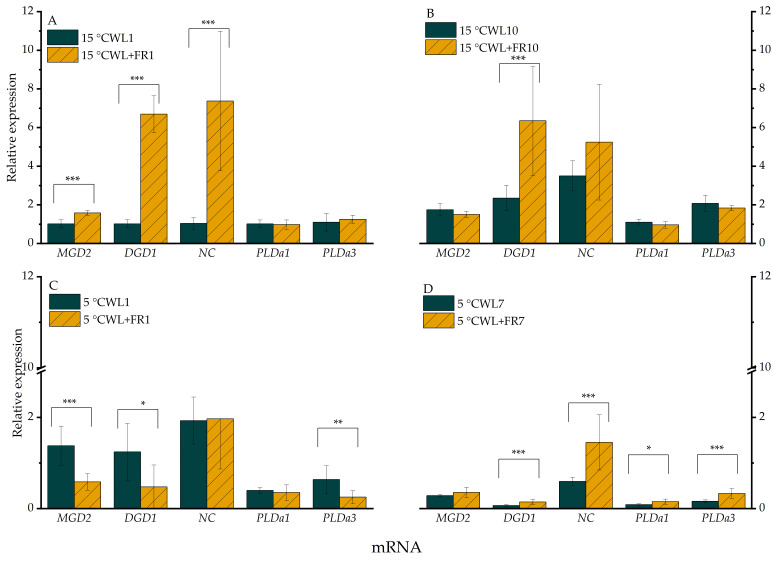
Relative expression levels of selected genes in leaves of barley “Nure” plants incubated at 15 °C for 1 day (**A**) and for 10 days (**B**), and leaves grown at 5 °C for 1 day (**C**) and for 7 days (**D**), under WL and WL+FR. Mean values ± SD of 9 independent biological replicates are shown. Significance levels are designated as * *p* < 0.1, ** *p* < 0.05, *** *p* < 0.01.

**Figure 3 ijms-21-07557-f003:**
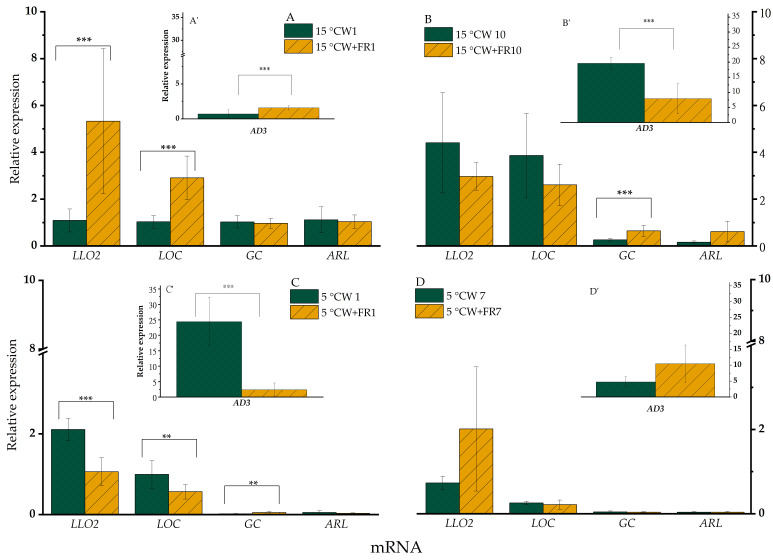
mRNA relative expression of barley “Nure” leaves grown at 15 °C for 1 day (**A**,**A’**) and for 10 days (**B**,**B’**), and leaves grown at 5 °C for 1 day (**C**,**C’**) and for 7 days (**D**,**D’**), under WL and WL+FR. Significance level was determined from 9 independent biological replicates and is designated as ** *p* < 0.05, *** *p* < 0.01.

**Figure 4 ijms-21-07557-f004:**
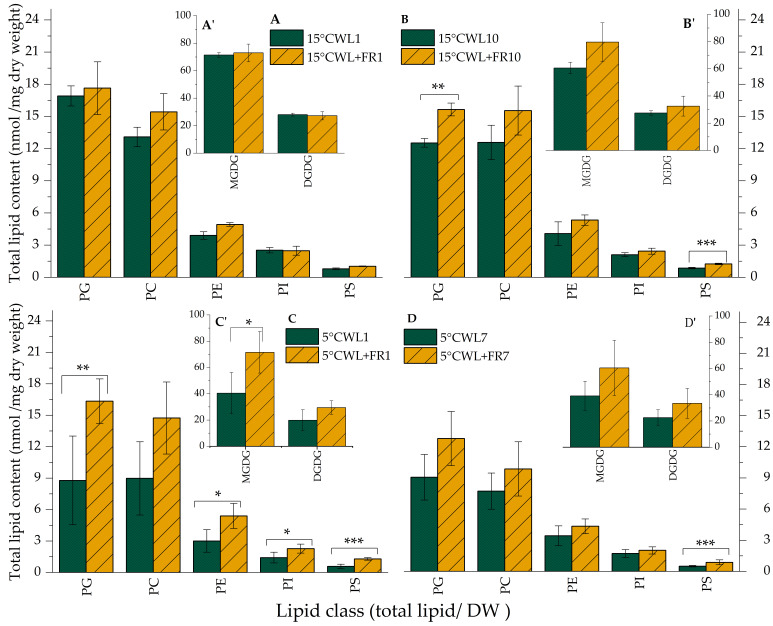
Lipid class distribution of barley “Nure” leaves grown at 15 °C for 1 day (**A**,**A’**) and for 10 days (**B**,**B’**), as well as in leaves grown at 5 °C for 1 day (**C**,**C’**) and for 7 days (**D**,**D’**) under WL and WL+FR. Mean values ± SD were calculated from 3 independent biological replicates. Significance levels are designated as * *p* < 0.1, ** *p* < 0.05, *** *p* < 0.01.

**Figure 5 ijms-21-07557-f005:**
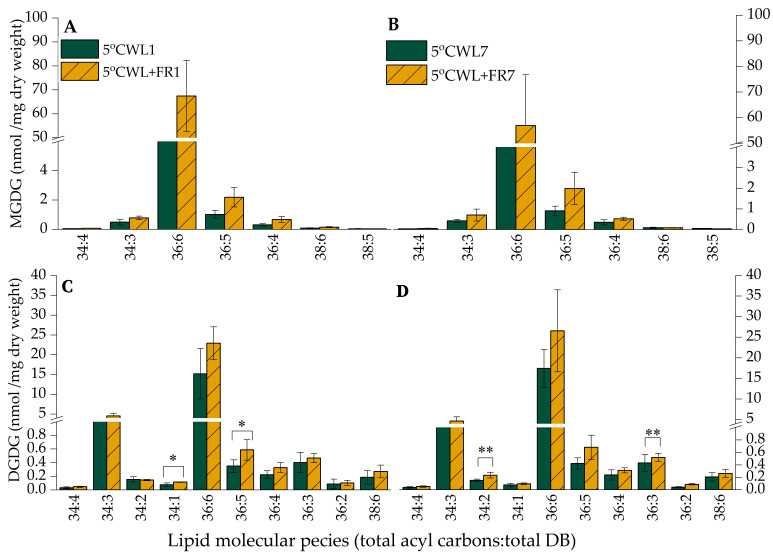
The amount of the most abundant MGDG and DGDG lipid species in barley “Nure” leaves grown at 5 °C for 1 day (**A**,**C**) and for 7 days (**B**,**D**) under WL and WL+FR. Lipid species are denoted as total carbon number: total double-bond. Mean values ± SD were calculated from 3 independent biological replicates. Significance levels are designated as * *p* < 0.1, ** *p* < 0.05.

**Figure 6 ijms-21-07557-f006:**
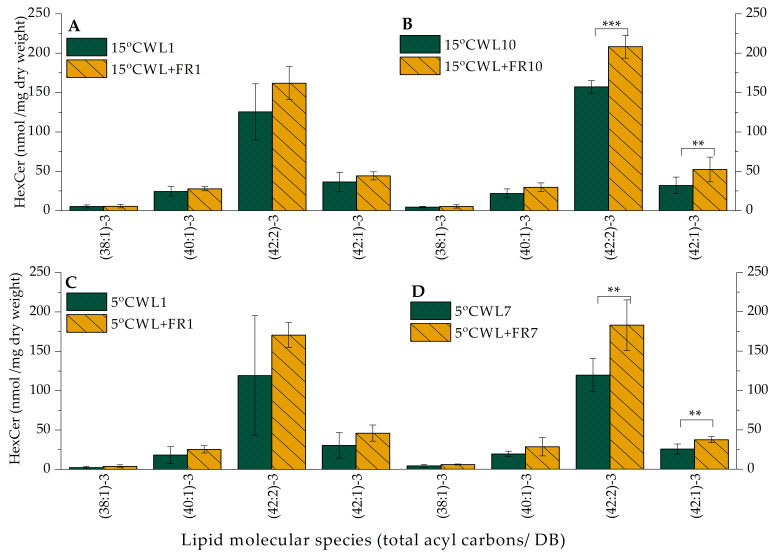
HexCer lipid-class distribution of barley “Nure” leaves grown at 15 °C for 1 day (**A**) and for 10 days (**B**), as well as in leaves grown at 5 °C for 1 day (**C**) and for 7 days (**D**), under WL and WL+FR. Mean values ± SD were calculated from 3 independent biological replicates. Significance levels are designated as ** *p* < 0.05, *** *p* < 0.01.

**Figure 7 ijms-21-07557-f007:**
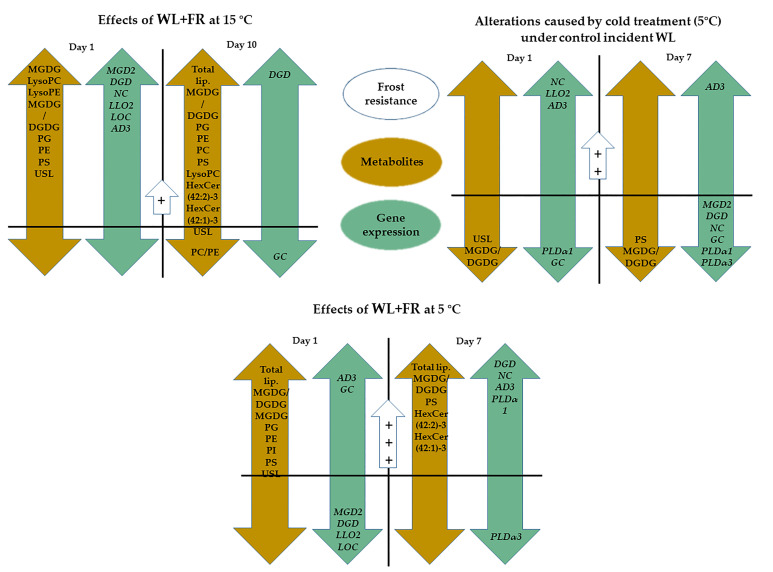
Our conclusion about the effect of WL-enriched FR and cold temperature on lipid content and quality of the expression of the related genes in barley leaves. Lipid composition changes under WL+FR: LysoPC, LyoPE, PC, PG, PE, PS. Lipid composition changes elicited by cold temperature: Total lipid, PI. The PC/PE lipid ratio decreased under WL+FR. Gene expression changes under WL+FR: *NC, DGD1, GC*. Gene expression changes elicited by cold temperature: *GC, NC, AD3,*
*PLDα1, PLDα3*. Gene expression changes elicited by both cold temperature and WL+FR: *GC, NC*. The decrease in *LLO2* and *LOC* gene expression positively influences membrane integrity after the cold acclimation period [71,72,73,74]. Symbols: Orange: Lipid metabolite changes; Green: Gene expression patterns; Blue: Frost resistance changes; The order of the induced frost tolerance: + < ++ < +++. The significance level in figures is *p* < 0.1.

**Table 1 ijms-21-07557-t001:** Total lipid content in barley leaves grown at 15 °C for 1 and 10 days and in leaves grown at 5 °C for 1 and 7 days under WL and WL+FR.

	Total Lipid		*p*	*p*
	WL (mol%)	WL+FR (mol%)		
15 °C 1 day	398.64 ± 57.02	473.52 ± 71.02	0.228	
15 °C 10 days	405.71 ± 62.59	549.34 ± 87.04	0.081	*
5 °C 1 day	292.3 ± 47.39	457.85 ± 70.17	0.028	**
5 °C 7 days	306.95 ± 47.09	472.46 ± 72.43	0.029	**

Mean ± SD values were calculated from three independent biological replicates. Significance levels in table are designated as * *p* < 0.1, ** *p* < 0.05. Mean ± SD values were calculated from 3 independent biological replicates.

**Table 2 ijms-21-07557-t002:** The MGDG/DGDG lipid ratios in barley leaves grown at 15 °C for 1 and 10 days and in leaves grown at 5 °C for 1 and 7 days under WL and WL+FR.

	MGDG/DGDG		*p*	*p*
	WL	WL+FR		
15 °C 1 day	2.54 ± 0.03	2.68 ± 0.08	0.047	**
15 °C 10 days	2.2 ± 0.07	2.45 ± 0.07	0.016	**
5 °C 1 day	2.04 ± 0.07	2.41 ± 0.12	0.01	***
5 °C 7 days	1.73 ± 0.04	1.8 ± 0.01	0.042	**

Significance levels in tables are designated as ** *p* < 0.05, *** *p* < 0.01. Mean ± SD values were calculated from 3 independent biological replicates.

**Table 3 ijms-21-07557-t003:** Total lipid double-bond index in barley leaves grown at 15 °C for 1 and 10 days and in leaves grown at 5 °C for 1 and 7 days under WL and WL+FR.

	DBI		*p*	*p*
	WL	WL+FR		
15 °C 1 day	4.78 ± 0.36	4.74 ± 0.23	0.879	
15 °C 10 days	4.8 ± 0.34	4.87 ± 0.5	0.851	
5 °C 1 day	4.78 ± 0.4	4.76 ± 0.37	0.952	
5 °C 7 days	4.6 ± 0.35	4.7 ± 0.73	0.841	

Double-bond indices were calculated as described in the Materials and Methods. Mean ± SD values were calculated from 3 independent biological replicates.

**Table 4 ijms-21-07557-t004:** PE species content in barley leaves grown at 15 °C for 1 and 10 days and in leaves grown at 5 °C for 1 and 7 days under WL and WL+FR.

	DB	PE		*p*	*p*
		WL	WL+FR		
15 °C 1 day	3	4.65 ± 0.67	5.33 ± 1.02	0.39	
	2	2.9 ± 0.06	3.37 ± 0.18	0.01	***
	6	1.67 ± 0.17	2.03 ± 0.39	0.21	
	5	2.79 ± 0.13	3.27 ± 0.29	0.06	**
	4	1.37 ± 0.21	1.67 ± 0.3	0.22	
15 °C 10 days	3	3.56 ± 0.6	5.36 ± 0.18	0.01	***
	2	3.31 ± 0.78	3.85 ± 0.56	0.39	
	6	1.02 ± 0.16	1.72 ± 0.3	0.02	**
	5	2.47 ± 0.42	3.46 ± 0.24	0.02	**
	4	1.77 ± 0.28	1.92 ± 0.26	0.51	
5 °C 1 day	3	2.8 ± 1.13	5.27 ± 1.01	0.05	**
	2	2.27 ± 0.7	4.23 ± 0.77	0.03	**
	6	0.9 ± 0.44	1.67 ± 0.45	0.11	
	5	1.97 ± 0.68	3.49 ± 0.68	0.05	**
	4	1.25 ± 0.37	2.13 ± 0.39	0.04	**
5 °C 7 days	3	2.9 ± 0.66	4.2 ± 1.7	0.29	
	2	3.04 ± 0.89	3.38 ± 0.14	0.54	
	6	0.85 ± 0.25	1.41 ± 0.9	0.35	
	5	2.17 ± 0.62	2.76 ± 0.78	0.36	
	4	1.6 ± 0.5	1.74 ± 0.07	0.65	

Amount of PE species was calculated as described in Materials and Methods. Significance levels in tables are designated as ** *p* < 0.05, *** *p* < 0.01. Mean ± SD values were calculated from 3 independent biological replicates.

**Table 5 ijms-21-07557-t005:** LysoPC in barley leaves grown at 15 °C for 1 and 10 days and in leaves grown at 5 °C for 1 and 7 days under WL and WL+FR.

	LysoPC (Signal/Mass Dry Tissue Weight)		*p*	*p*
	WL	WL+FR		
15 °C 1 day	0.37 ± 0.2	0.7 ± 0.08	0.057	*
15 °C 10 day	0.57 ± 0.42	0.46 ± 0.21	0.706	
5 °C 1 day	0.5 ± 0.29	0.66 ± 0.33	0.562	
5 °C 7 day	0.46 ± 0.32	0.49 ± 0.27	0.907	

Significance levels in tables are designated as * *p* < 0.1. Mean ± SD values were calculated from 3 independent biological replicates.

**Table 6 ijms-21-07557-t006:** Lyso PE in barley leaves grown at 15 °C for 1 and 10 days and in leaves grown at 5 °C for 1 and 7 days under WL and WL+FR.

	Lyso PE (Signal/Mass Dry Tissue Weight)		*p*	*p*
	WL	WL+FR		
15 °C 1 day	0.25 ± 0.05	0.39 ± 0.03	0.014	**
15 °C 10 day	0.3 ± 0.14	0.36 ± 0.16	0.651	
5 °C 1 day	0.37 ± 0.12	0.43 ± 0.12	0.573	
5 °C 7 day	0.35 ± 0.1	0.4 ± 0.05	0.482	

Significance levels in tables are designated as ** *p* < 0.05. Mean ± SD values were calculated from 3 independent biological replicates.

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
