# Peer review of "Decreased R:FR Ratio in Incident White Light Affects the Composition of Barley Leaf Lipidome and Freezing Tolerance in a Temperature-Dependent Manner"

_ijms, 2020, doi:10.3390/ijms21207557_

Round 1
Reviewer 1 Report
In this manuscript, the authors aimed to study the changes in the lipidome of barley leaves when treated with complementary far-red (FR) illumination at different temperatures. They concluded that white light (WL) + FR can substitute cold treatment at higher temperature (15°C), and mostly acts in a synergistic manner at cold (5°C) nonfreezing temperature. I have some concerns with the rationale, experimental design, and conclusions.
- I want to understand why the authors decided to study the impact of FR on the composition of barley leaf lipidome and freezing tolerance. Throughout the manuscript, the authors treated barley with WL or WL+FR under 12-h day length. Under natural conditions, low R:FR conditions are observed in the vegetative canopy (shade) or during the twilight period. Did the authors try to mimic shade or dusk? If the latter, an end-of-day (EOD) FR treatment would be more suitable for this study.
- Because of the above-mentioned confusion, I don’t think the introduction was accurate enough in providing related background information to this study. At least, it was lengthy and not well organized.
- I don’t think the expression analysis in Fig. 2 & 3 was carefully designed. I noticed that the authors collected samples after 6-8 h illumination so I think they understand the circadian gating effect on the expression of many genes. I’m wondering why they chose this timing (ZT6 to ZT8) to study the transcript levels of selected genes. This needs to be explained.
- I also didn’t see a good correlation between the gene expression data and lipid content (e.g. MGD2 and DGD1 in Fig.2 vs. MGDG and DGDG in Fig. 4).
- Some descriptions in the Results were not accurate.
- Line 249: “Upon WL+FR treatment the MGDG content in barley leaves was greater at 5°C and 15°C after 1 day.” I don’t see this in Fig.4A’.
- Line 254: “PE content was significantly greater only after 1 day of WL+FR treatment at 15°C, …” I don’t see this in Fig.4A.
- A bigger question I have about the lipid contents shown in Fig.4 is that I don’t see big enough changes (fold-wise) in most of the comparisons between WL and WL+FR, although some of them did show statistical significance. I’m not convinced that the phenotype (lipid content) was obvious enough to draw some conclusions in the manuscript. For example, the authors mentioned in line 353-354 that “WL+FR “markedly” elevated the PG and PS levels of barley leaves after 10 days at 15°C as compared to the WL control, …” I don’t agree. The difference was really tiny to me, so I think it was overstated. The authors should tone down some of their conclusions.
- The discussion part is also lengthy and not well organized. The authors should remove unrelated information and focus on summarizing their observations and draw meaningful conclusions. For example, I don’t think line 336-349 is necessary and can be removed.
Author Response
Q1
I want to understand why the authors decided to study the impact of FR on the composition of barley leaf lipidome and freezing tolerance. Throughout the manuscript, the authors treated barley with WL or WL+FR under 12-h day length. Under natural conditions, low R:FR conditions are observed in the vegetative canopy (shade) or during the twilight period. Did the authors try to mimic shade or dusk? If the latter, an end-of-day (EOD) FR treatment would be more suitable for this study.
A1.
In our previous study (Novák et al., 2016.) we investigated the effect of the complementary FR light both in case of end-of-day and also whole day settings on gene expression, photosynthesis efficiency, and freezing tolerance as well. The obtained results showed a greater influence on gene expressions and freezing tolerance due to whole day FR supplementation. The results of end-of-day treatments were similar but not as prominent. The whole day treatment was selected because the effect of the supplementary FR light on the plant lipidome (which may eventually lead a significant improvement in the freezing tolerance of plants) was the focus of the present study.
Novák A, Boldizsár Á, Ádám É, Kozma-Bognár L, Majláth I, Båga M, Tóth B, Chibbar R, Galiba G. Light-quality and temperature-dependent CBF14 gene expression modulates freezing tolerance in cereals. J Exp Bot. 2016 Mar;67(5):1285-95. doi: 10.1093/jxb/erv526. Epub 2015 Dec 27. PMID: 26712822.
Q2
Because of the above-mentioned confusion, I don’t think the introduction was accurate enough in providing related background information to this study. At least, it was lengthy and not well organized.
A2 We reorganized the introduction, moreover deleted parts that are not closely related to the topic. A sentence was also added to elucidate why we used ZT6 to ZT8 for sampling see Q3:’ It is also well established that some of the CBF-genes are circadian regulated and their expression reaches a maximum after 6-12 h after dawn in wheat and barley, but the FR light causes earlier transcription (ZT4 to ZT8) [6, 9].’
Q3.
I don’t think the expression analysis in Fig. 2 & 3 was carefully designed. I noticed that the authors collected samples after 6-8 h illumination so I think they understand the circadian gating effect on the expression of many genes. I’m wondering why they chose this timing (ZT6 to ZT8) to study the transcript levels of selected genes. This needs to be explained.
A3.
Our main goal was to elucidate in which way the plant lipidome responds to FR light which eventually leads to significant improvement in the freezing tolerance of plants. In case of freezing tolerance, it is important to analyze the expression of CBFs and its downstream genes. Since FR light enhances cold hardening process in plants and low R/FR ratio increases the expression of the CBFs and COR genes in A. thaliana (Franklin et al. 2007, Ahres et al.,2020). The CBF gene family is gated by the circadian clock with the peak of each rhythm occurring 6–12 h after subjective dawn in wheat (Badawi et al.,2007). In our previous study, it was found that the peak of many CBF genes occurred 4–12 h after dawn in barley, but the FR light causes earlier transcription (ZT4 to ZT8) (Gierczik et al. 2017). This phenomenon was observed in case of the HvCBF4-subgroup members of CBF gene family which mainly influence the freezing tolerance of cereals. We decided the timing to take samples accordingly.
Franklin, K.A.; Whitelam, G.C. Light-quality regulation of freezing tolerance in Arabidopsis thaliana. Nat. Genet.2007,39, 1410–1413.
Ahres, M.; Gierczik, K.; Boldizsár, Á.; Vítámvás, P.; Galiba, G. Temperature and Light-Quality-Dependent Regulation of Freezing Tolerance in Barley. Plants 2020, 9, 83.
Badawi M, Danyluk J, Boucho B, Houde M, Sarhan F. The CBF gene family in hexaploid wheat and its relationship to the phylogenetic complexity of cereal CBFs. Mol Genet Genomics. 2007 May;277(5):533-54. doi: 10.1007/s00438-006-0206-9. Epub 2007 Feb 7. PMID: 17285309; PMCID: PMC2491707.
Gierczik, K.; Novák, A.; Ahres, M.; Székely, A.; Soltész, A.; Boldizsár, Á.; Gulyás, Z.; Kalapos, B.; Monostori, I.; Kozma-Bognár, L.; Galiba, G.; Vágújfalvi, A. Circadian and Light Regulated Expression of CBFs and their Upstream Signalling Genes in Barley. Int. J. Mol. Sci. 2017, 18, 1828.
Q4
I also didn’t see a good correlation between the gene expression data and lipid content (e.g. MGD2 and DGD1 in Fig.2 vs. MGDG and DGDG in Fig. 4).
A4
We redrafted the description of the gene expression and lipid content and described the relations between them more carefully see: Page 10.
Q5
- Some descriptions in the Results were not accurate.
- Line 249: “Upon WL+FR treatment the MGDG content in barley leaves was greater at 5°C and 15°C after 1 day.” I don’t see this in Fig.4A’.
- Line 254: “PE content was significantly greater only after 1 day of WL+FR treatment at 15°C, …” I don’t see this in Fig.4A.
A5
We redrafted the Results and corrected the marked points.
Q6
A bigger question I have about the lipid contents shown in Fig.4 is that I don’t see big enough changes (fold-wise) in most of the comparisons between WL and WL+FR, although some of them did show statistical significance. I’m not convinced that the phenotype (lipid content) was obvious enough to draw some conclusions in the manuscript. For example, the authors mentioned in line 353-354 that “WL+FR “markedly” elevated the PG and PS levels of barley leaves after 10 days at 15°C as compared to the WL control, …” I don’t agree. The difference was really tiny to me, so I think it was overstated. The authors should tone down some of their conclusions.
A6
We think that the lipid composition and ratios cannot change in a wide range without disturbing the membrane integrity and function. Genetic modification can induce large variations of lipid ratios but these plants are severely affected they membrane functions have defected. For example, MGDG level below 10 % can be lethal. We expect only moderate changes under physiological conditions during acclimation to environmental changes. The modulation of membrane properties does not require the change of bulk lipid content. The changes we observed have a similar extent as can be found in the literature. (see below).
Nevertheless, we restated our conclusions to make our claims more moderate.
Welti, W. Li, M. Li, Y. Sang, H. Biesiada, H.E. Zhou, C.B. Rajashekar, T.D. Williams, X. Wang, Profiling membrane lipids in plant stress responses. Role of phospholipase D alpha in freezing-induced lipid changes in Arabidopsis, The Journal of biological chemistry, 277 (2002) 31994-32002.
Welti, J. Shah, W. Li, M. Li, J. Chen, J.J. Burke, M.L. Fauconnier, K. Chapman, M.L. Chye, X. Wang, Plant lipidomics: discerning biological function by profiling plant complex lipids using mass spectrometry, Frontiers in bioscience : a journal and virtual library, 12 (2007) 2494-2506.
Q7
The discussion part is also lengthy and not well organized. The authors should remove unrelated information and focus on summarizing their observations and draw meaningful conclusions. For example, I don’t think line 336-349 is necessary and can be removed.
A7
According to the request of the reviewer, the discussion part was reorganized. The unrelated information was removed and the concluding part was re-written carefully. Please, consider the modified ‘Discussion’.
Reviewer 2 Report
Dear author(s)
Thanks a lot for your work. Except for the comment to the statistical analysis in M&M, I only have suggestions for minor/style issues.
- If possible, shorten sentences for better reading.
- Use no-break spaces between numbers and units.
- Stay cohesive with spacing when using e.g. ">" or "±".
- E.g. in line 184 "p <0.05" and in line 211 "p<0.05"
- Check with journal guidelines, but in general, it should be a space on both sides.
- Please increase the font size in figures.
- Please broaden the lines in figures.
- Please give more details for statistical analysis, e.g. model etc.
Author Response
Q1-6
- If possible, shorten sentences for better reading.
- Use no-break spaces between numbers and units.
- Stay cohesive with spacing when using e.g. ">" or "±".
- E.g. in line 184 "p <0.05" and in line 211 "p<0.05"
- Check with journal guidelines, but in general, it should be a space on both sides.
- Please increase the font size in figures.
- Please broaden the lines in figures.
- Please give more details for statistical analysis, e.g. model etc.
A1-5
We’ve checked and rephrased the long sentences throughout the manuscript and modified the text and figures according to the suggestions.
A6
A short section was inserted describing the statistical methods used in this study. Please see ‘4.8. Statistics and reproducibility’ in the Materials and Methods.
Round 2
Reviewer 1 Report
The authors provided reasonable answers to most of my questions. I do think that most experiments were carefully designed and well performed. Thus, this is a technically sound paper. However, the data were not impressive. This manuscript focused on the changes in leaf lipidome. I want to emphasize that I don't think the changes in lipid contents were significant enough. I'm glad to see that the authors toned down some of their conclusions. Please change the following statements as well because they are not accurate based on the data shown in Fig. 4 & Fig. 5.
- Line 247-248: "Upon WL+FR treatment the MGDG content in barley leaves was greater at 5 °C after 1 and 7 days, and 15 °C only after 10 days." I don't see the statistical analysis for MGDG content at 15 °C after 10 days (Fig. 4B'). Does this mean no significant difference between WL and WL+FR? If so, the description was not accurate.
- Line 252: "PE content was greater after 1 day of WL+FR treatment at 15 °C, ..." I don't see the statistical analysis for PE content at 15 °C after 1 or 10 days (Fig. 4A & 4B). Does this mean no significant difference between WL and WL+FR? If so, the description was not accurate.
- Line 275-276: "..., an increase in the amount of each lipid species (most considerable in the amount of 36:6 molecular species) was noticeable at 5 °C (Fig. 5A and B)" I don't see the statistical analysis for lipid content at 5 °C after 1 or 7 days (Fig. 5A & 5B). Does this mean no significant difference between WL and WL+FR? If so, the description was not accurate.
Author Response
A1
Thank You for carefully reviewing our manuscript. We corrected the sentence and only the treatment causing significant effect remained in the text: Page 7 line 248 ..’ significantly greater at 5 °C after 1 day.‘…
A2
The following correction was made on Page 7 lines 252-253: ..‘PE content was significantly greater after 1 day of WL+FR treatment at 5 °C, whereas PG increased significantly both after 1 day at 5 °C and after 10 day at 15 °C (Fig. 4A, B and C).’ ..
A3
The accurate description of this part is the following: Page 9 lines 274-278, ..’Although there was no significant alteration within the MGDG lipid molecular class in barley leaves upon WL+FR treatment at 15 °C (Supplementary Fig. 4), an increase in the amount of each lipid species was noticeable at 5 °C (Fig 5A and B). In the case of the DGDG molecular species WL+FR increased the content of 34:1, 36:5 and 34: 2, 36: 3 lipid species in the barley leaves at 5 °C after the 1 and 7-day treatment (Fig 5C and D).’..